# Korean Nationwide Exploration of Sarcopenia Prevalence and Risk Factors in Late Middle-Aged Women

**DOI:** 10.3390/healthcare12030362

**Published:** 2024-01-31

**Authors:** Jongseok Hwang, Soonjee Park

**Affiliations:** 1Institute of Human Ecology, Yeungnam University, Gyeongsan-si 38541, Republic of Korea; sfcsfc44@naver.com; 2Department of Clothing and Fashion, Yeungnam University, Gyeongsan-si 38541, Republic of Korea

**Keywords:** sarcopenia, prevalence, risk factor, odds ratio, Korean nationwide study

## Abstract

This study examined specific clinical risk factors for age-related loss of skeletal muscle mass in late middle-aged women with sarcopenia. This Korean nationwide cross-sectional study analyzed data from 2814 community-dwelling women aged from 50 to 64 years old and screened them for sarcopenia. This study examined various risk factors such as age; height; weight; body mass index; waist circumference; skeletal muscle mass index; systolic and diastolic blood pressure; smoking and drinking habits; fasting glucose levels; triglyceride; and cholesterol levels. Complex sampling analysis was used for the data set. Prevalence of sarcopenia with a weighted prevalence of 13.43% (95% confidence interval: 2.15–15.78). The risk factors for sarcopenia were height, body mass index, waist circumference, skeletal muscle mass index, systolic blood pressure, diastolic blood pressure, triglyceride level, and total cholesterol level (*p* < 0.05). Weight, fasting glucose level, drinking status, and smoking status were not significant (*p* > 0.05). These results are expected to contribute to the existing literature on sarcopenia and identify potential risk factors associated with the development of sarcopenia in late middle-aged females. By acknowledging prevalence and recognized risk factors, healthcare professionals may augment their proficiency in recognizing and discerning potential instances of sarcopenia in female patients.

## 1. Introduction

Sarcopenia is identified via age-related declines in skeletal muscle mass, leading to progressive and generalized skeletal muscle disorder [1]. Although various studies have indicated that hormonal changes, immobility, age-related changes in muscle composition, nutritional factors, and neurodegenerative processes contribute to its development, the precise mechanism of sarcopenia is not yet fully understood. Importantly, sarcopenia is particularly common among individuals aged 65 years and above [2].

The elderly population in Asia is rapidly growing, and Korea is among the countries with the highest rates of population aging worldwide. As of 2021, approximately 16.5% of the Korean population was aged 65 years or older, and this percentage is expected to increase to approximately 40% by 2050 [3]. Sarcopenia is more prevalent in Korea and Asia than in other countries.

Several studies have reported a higher prevalence of sarcopenia in females compared to males. In a screen of 10,063 individuals, Dam et al. announced a prevalence of 11.80% in females and 5.10% in males [4]. Similarly, Hunt et al. examined an older population of about two thousand Japanese individuals in a community dwelling and found a sarcopenia prevalence of 16.56% in females and 10.34% in males [5]. A significant proportion of the elderly population in Korea, particularly women, is vulnerable to sarcopenia.

However, compared with the extensive research on sarcopenia in males, the early detection of sarcopenia in females remains a challenging task [6,7,8,9]. Healthcare professionals, encompassing physical therapists and primary care clinicians, face difficulties in diagnosing sarcopenia owing to their limited knowledge and diagnostic tools despite the potential negative consequences associated with sarcopenia and the growing population of elderly females. In primary clinical settings, clinicians must assess the likelihood of sarcopenia before considering referrals for diagnosis and treatment. Moreover, the lack of awareness among clinicians regarding sarcopenia as a distinct disease increases the risk of missed diagnoses [10]. Therefore, it is crucial for healthcare professionals and primary clinicians to understand the characteristics of key risk factors associated with early detection and prevention to effectively address this challenge [11]. Prompt identification and early detection of individuals manifesting symptoms indicative of sarcopenia are essential to ensure timely diagnosis and intervention. Failure to diagnose the condition promptly can result in complications. However, the majority of studies on sarcopenia have primarily focused on individuals aged 65 years and older [12,13,14,15], even though age-related muscle loss can begin as early as the 50s [16,17,18,19,20]. It is crucial to identify the risk factors for muscle loss at an earlier stage to effectively prevent and treat this condition. Therefore, this study aimed to investigate the prevalence of sarcopenia and its associated risk factors in women aged 50 to 64 years. We hypothesized that this age group would have specific risk factors and prevalence rates that would differ from those observed in older individuals.

## 2. Methods

### 2.1. Study Population

The Korean National Health and Nutrition Examination Survey (KNHANES) was designed to investigate the health and nutritional status of non-institutionalized individuals in South Korea. The program was conducted by the Disease Control and Prevention Center. The KNHANES experimental procedures were approved by the Disease Control and Prevention Center Ethics Review Board and all participants signed and agreed to a written informed consent form. The ethics review board screened for human subjects’ ethical considerations, reviewed aspects such as the research plan, the informed consent process, participant safety, potential conflicts of interest, and the protection of personal information. The KNHANES IRB ensured that the research was lawful, ethical, and safeguarded the rights and safety of participants.

For eligibility in the sarcopenia group, participants had to satisfy three criteria: (1) be female, (2) have an age range of 50 to 64 years, and (3) to be within range of diagnosis of sarcopenia. Conversely, the normal group comprised women participants meeting the following a criterium: (1) aged between 50 and 64 years. Exclusions encompassed (1) pregnant individuals and (2) those who had undergone a diagnostic procedure involving contrast agent use in the week preceding the survey.

A total of 37,753 participants participated in the 2008–2011 KNHANES survey. Of these, 34,123 individuals were excluded due to being males or below 50 or above 64 years of age, and the remaining 4087 were female. Another 1273 subjects were excluded because they did not undergo health surveys or dual X-ray absorptiometry procedures. The number of female participants included in the final analysis set was 2814. The participants were divided into two groups based on the sarcopenia criteria, with 378 subjects in the sarcopenia group and 2436 in the normal group.

### 2.2. Variables

The study encompassed multiple variables for analysis, including age, height (in cm), weight (in kg), body mass index (BMI), waist circumference (WC), skeletal muscle index (SMI), smoking and drinking status, fasting glucose, triglycerides, total cholesterol (TC), systolic blood pressure, and diastolic blood pressure.

To measure WC, the circumference was determined at the midpoint between the lower rib cage and the upper edge of the iliac crest while the participant was exhaling fully.

Blood analyses were performed following an eight-hour fasting interval, and measurements of systolic and diastolic blood pressures were obtained utilizing a mercury sphygmomanometer after a ten minute repose in a seated posture. This meticulous approach to data collection ensured the accuracy and reliability of the physiological parameters under investigation.

The fasting period of eight hours, during which participants abstained from food intake, is a standard practice aimed at obtaining baseline blood values unaffected by recent meals. This allowed for a more accurate assessment of fasting glucose levels and other metabolic markers, providing insights into the participants’ physiological status.

Additionally, the choice of utilizing a mercury sphygmomanometer for the measurement of systolic and diastolic blood pressures underscored the commitment to precision in the study. The sphygmomanometer, a traditional and widely accepted instrument for blood pressure measurement, is known for its accuracy. The readings obtained using this device are considered reliable indicators of cardiovascular health, contributing to the robustness of the study’s findings.

The ten-minute rest period in a seated position before blood pressure measurements served multiple purposes. Firstly, it allowed participants to achieve a stable physiological state, minimizing the potential influence of transient factors on blood pressure readings. Secondly, the seated position standardized the conditions, ensuring uniformity across participants and reducing confounding variables that could compromise the internal validity of the study.

The assessment of smoking and drinking habits involved classifying participants into distinct categories based on their usage patterns. Participants were stratified into three groups: non-users, ex-users, or current users, providing a nuanced understanding of their tobacco and alcohol consumption behaviors.

This categorization approach was crucial for delving deeper into the complexities of smoking and drinking statuses within the study cohort. “Non-users” denoted individuals who had never engaged in smoking or drinking, establishing a baseline for those with no history of tobacco or alcohol consumption. “Ex-users” referred to individuals who were previously involved in smoking or drinking but had since ceased these behaviors. This category recognized the dynamic nature of lifestyle choices and allowed for the examination of potential long-term effects even after discontinuation. Lastly, “current users” represented individuals actively engaged in smoking or drinking at the time of the study, shedding light on immediate associations between these behaviors and other risk factors under investigation.

Moreover, the inclusion of these detailed categories offered a more nuanced exploration of the interplay between smoking, drinking, and various health parameters. Understanding the distinct characteristics of each subgroup enables researchers to discern potential trends, associations, or contrasts in health outcomes. For instance, comparing the health profiles of non-users, ex-users, and current users can illuminate whether the cessation of smoking or drinking is associated with specific health improvements or challenges. These variables were included in the analyses.

### 2.3. Criteria for Sarcopenia

The identification of sarcopenia, classified under the ICD-10-CM code M62.84, necessitates a meticulous evaluation of skeletal muscle mass in the extremities. In this study, the quantification of skeletal muscle mass in the limbs was conducted using dual X-ray absorptiometry (DXA), employing QDR4500A equipment from Hologic, Inc., Bedford, MA, USA. This technologically advanced method ensured precise and reliable measurements for an accurate assessment of skeletal muscle mass.

To gauge muscle mass effectively, the study employed the ASM (kg)/BMI (kg/m^2^) ratio, commonly referred to as the skeletal muscle mass index (SMI). The SMI calculation offers a quantitative representation of the relationship between muscle mass and body mass, allowing for a more nuanced understanding of muscular health. This index is particularly valuable in the context of sarcopenia diagnosis.

The diagnostic threshold for sarcopenia in women was established based on the SMI value. According to the criteria stipulated by the Foundation for the National Institutes of Health Sarcopenia Project [21], a diagnosis of sarcopenia was confirmed when the SMI value fell below 0.521. This criterion provided a standardized and objective measure for identifying individuals with insufficient skeletal muscle mass relative to their body mass, aligning with established norms in the field.

The robustness of the diagnostic methodology employed in this study underscored its reliability in accurately identifying sarcopenia among the study participants. By adhering to established criteria and utilizing advanced DXA technology, the research not only contributes to the scientific understanding of sarcopenia but also sets a benchmark for accurate and consistent diagnostic practices. The utilization of specific equipment and adherence to standardized criteria enhance the credibility of the study’s findings, ensuring that the diagnosis of sarcopenia was grounded in rigorous scientific methodology. This meticulous approach has established a foundation for further research and clinical applications related to sarcopenia and its diagnostic parameters.

The validity and reliability of DEXA are well-established [22,23,24].

### 2.4. Data Analysis

This study used the mean and standard deviation as statistical measures to summarize the data for each measurement. To ensure a representative analysis at the national level in Korea, complex sampling analysis was employed, incorporating the individual weights provided by the KNHNES. The statistical analyses were conducted through the utilization of SPSS software (version 22.0; IBM Corporation, Armonk, NY, USA).

The Information presented in this study revealed that the data employed adhered to a stratified, clustered, and multistage probability sampling design. To examine the variations in chemical parameters between participants afflicted with sarcopenia and those without, independent *t*-tests and chi-square analyses were employed. This meticulous sampling approach ensured a comprehensive and representative selection, allowing for a nuanced exploration of the chemical profiles in relation to sarcopenia. The statistical tools utilized independent *t*-tests and chi-square analyses. Multiple logistic regression analysis was used to calculate the odds ratio for sarcopenia. The statistical significance level was set at *p* = 0.05, to determine the presence of statistically significant associations.

## 3. Results

### 3.1. Prevalence

Table 1 illustrates the prevalence of sarcopenia, with a weighted prevalence of 13.43% (95% confidence interval (CI): 2.15–15.78) (Figure 1). The unweighted prevalence stood at 13.87%, while the remaining percentage represents the normal group, amounting to 86.57%.

### 3.2. Clinical Risk Factors

Height, BMI, WC, SMI, TC level, triglyceride level, systolic blood pressure, and diastolic blood pressure were significantly different between the two groups (*p* < 0.05). Weight, fasting status, smoking status, and drinking variables were not significantly different between the groups (*p* > 0.05) (Table 2).

### 3.3. Multiple Logistic Regression for Odd Ratio

Table 3 shows the odds ratios (oRs) with 95% confidence intervals using logistic regression analysis. BMI 19.178 (4.3080–85.380), WC 1.362 (1.011–1.836), SMI 0.115 (0.067–0.193), SBP 1.349 (1.255–1.451), DBP 1.845 (1.646–2.067), TC 1.166 (1.141–1.191), and Triglyceride 1.078 (1.058–1.097) were statistically significant.

## 4. Discussion

This study aimed to evaluate the prevalence and risk factors associated with sarcopenia among community dwelling late middle-aged females. Aging populations in Korea and Asia are rapidly increasing, leading to a higher occurrence of sarcopenia, especially among females. Despite the potential adverse effects of sarcopenia, healthcare professionals face challenges in diagnosing the condition due to a lack of adequate knowledge and diagnostic tools, resulting in overlooked diagnoses and complications. It is useful to utilize variables such as age, height, weight, BMI, WC, SMI, smoking and drinking status, SBP, DBP, fasting glucose levels, triglyceride, and TC. The above variables are a cost-effective, convenient, and accessible approach for identifying patients with potential sarcopenia. Recognizing the risk factors is essential for the early detection and prevention of sarcopenia. The identified risk factors in females included measurements such as WC, SMI, SBP and DBP triglyceride level, and TC.

The consistent identification of waist circumference as a risk factor for sarcopenia has been a focal point in various female sarcopenic studies [25,26,27]. One American study revealed sarcopenic individuals had an enlarged waist circumference [25]. Likewise, a Brazilian cohort study reported a greater waist circumference among individuals with sarcopenia than those without sarcopenia [26]. A separate investigation conducted in Japan revealed that those identified with sarcopenia demonstrated greater waist circumferences compared to a heathy population [28].

The theoretical underpinning of the observed increase in waist circumference in adults with sarcopenia is rooted in the interconnected relationship between increased fat mass and diminished muscle mass [28]. Individuals with sarcopenia often have problems with muscle power and function due to muscle loss, resulting in decreased engagement in physical activities, such as challenges in sit-to-stand and walking extended distances both indoors and outdoors [29]. This decline in physical activity is strongly correlated with a reduction in total daily energy expenditure and an increase in body fat stores. In particular, it accumulates in the visceral and abdominal regions, ultimately leading to the expansion of waist volume [29]. Consequently, the correlation between diminished muscle mass and fat mass accumulation in sarcopenia is bidirectional and reinforces this hypothesis [30]. Thus, evidence consistently highlighting waist circumference as a discernible risk factor for sarcopenia emphasizes the need for a nuanced understanding of the intricate interplay between muscle and fat mass.

The other identified risk factor in the blood laboratory examination was an elevation in triglyceride level, a finding that was consistent with previous investigations [31,32,33]. In a cross-sectional study conducted by Lu et al. [33] that was conducted on subjects from east China, it was observed that females presenting with sarcopenia manifested heightened serum triglyceride levels. Similarly, in their examination of an older adult cohort in the northern region of Taiwan, Lu et al. discerned a notable increase in triglyceride levels within a demographic characterized by sarcopenia. Correspondingly, Buchmann et al. [31], through their examination of an elderly population in Berlin, concluded that triglyceride levels were elevated within the subset afflicted with sarcopenia compared to their non-sarcopenic counterparts. This collective body of evidence underscores the consistency of the association between sarcopenia and elevated triglyceride levels across diverse geographic and demographic spectra, thereby reinforcing the robustness of the observed correlation.

Insulin resistance is a plausible underlying mechanism for the observed correlation between sarcopenia and elevated triglyceride levels. Insulin resistance disrupts lipid metabolism. Under normal conditions, insulin facilitates the uptake of fatty acids and glucose by adipose tissue. In insulin resistance, this regulatory process is impaired, resulting in an increased release of fatty acids from adipose tissue into the bloodstream [34]. Skeletal muscle is a pivotal primary repository, storing approximately 80% of ingested glucose after meals, thereby acting as a critical prevention of hyperglycemia in the bloodstream [19]. However, individuals with sarcopenia, particularly women, frequently exhibit a notable reduction in insulin sensitivity. This lowered insulin sensitivity displays a diminished capacity for glucose uptake by skeletal muscles, stemming from lower proportions of type I muscle fibers and a reduced capillary density susceptible to insulin action [35]. Furthermore, the accumulation of fat in sarcopenic adults, as mentioned in the discussion on waist circumference, contributes to the synthesis of triglycerides, facilitated by the liver through lipogenesis, and the liver’s orchestration of triglyceride synthesis via lipogenesis [36]. The liver, a central organ in the intricate lipid metabolism network, demonstrates a discerning reaction when confronted with excess circulating fatty acids in a systemic environment. This response is characterized by the initiation of lipogenesis. Within this intricate biochemical pathway, the liver engages in triglyceride synthesis from an abundance of fatty acids and glycerol molecules, representing a pivotal juncture in the overall metabolic panorama. Lipogenesis, at its core, embodies a molecular performance orchestrated within hepatic cells. This intricate choreography of enzymatic reactions unfolds within hepatocytes and culminates in the conversion of fundamental building blocks, fatty acids, and glycerol molecules into more intricate and storage-ready triglycerides. These enzymatic transformations transcend biochemical processes. Instead, they reflect the meticulously regulated finesse of the liver’s molecular machinery, with each enzymatic step intricately controlled to ensure seamless synthesis of triglycerides [37,38].

Total cholesterol was recognized as a risk factor for sarcopenia, which was consistent with the results of previous studies [27,32]. According to Du et al. [32], females with sarcopenia exhibit elevated total cholesterol levels compared to their counterparts in the normal group. Similarly, Sanada et al. [27] assessed a Japanese population and observed significantly higher total cholesterol levels in individuals diagnosed with sarcopenia than those in the normal group.

The potential cause for elevated total cholesterol levels may be attributed to both inflammation and mitochondrial dysfunction. Aging is characterized by chronic low-grade inflammation, commonly referred to as “inflammaging”, which plays a role in muscle wasting by facilitating the breakdown of muscle proteins and hindering the regeneration of muscle tissue [39]. Furthermore, the intricate relationship between sarcopenia and mitochondrial dysfunction contributes to our understanding of the mechanisms behind altered cholesterol levels [40]. Mitochondria, the cellular powerhouses responsible for energy production, undergo changes with aging that can lead to diminished energy output and subsequent muscle fatigue.

To delve deeper into the relationship between total cholesterol and sarcopenia, it is crucial to explore the specific mechanisms by which inflammation and mitochondrial dysfunction influence lipid metabolism and muscle health. Inflammatory mediators, such as cytokines, can impact the liver’s synthesis of lipoproteins, including cholesterol-carrying molecules [41]. This alteration in lipid metabolism may contribute to the observed elevation in total cholesterol levels in individuals with sarcopenia. Mitochondrial dysfunction, on the other hand, can affect the energy balance within muscle cells. The decline in mitochondrial function diminishes the efficiency of energy production, potentially influencing the regulation of lipid metabolism [42]. This not only contributes to muscle fatigue, a characteristic of sarcopenia, but may also play a role in the dysregulation of cholesterol levels. Moreover, underscoring the importance of considering hormonal influences and metabolic differences in the interplay between cholesterol and muscle health [43]. Hormonal changes associated with aging, including alterations in growth hormone and estrogen hormones, can impact both lipid metabolism and muscle mass maintenance. Understanding the intricate connections between total cholesterol, inflammation, mitochondrial dysfunction, and sarcopenia necessitates a comprehensive examination of cellular and molecular processes. Research in this area holds promise not only for elucidating the pathophysiology of sarcopenia but also for identifying potential targets for therapeutic interventions aimed at mitigating muscle loss and optimizing metabolic health.

Our study’s findings demonstrated that systolic and diastolic blood pressures serve as risk factors for women, which was consistent with prior research [33,44]. An investigation conducted in Taiwan by Lu et al. revealed that individuals within the sarcopenia group exhibited elevated SBP and DBP compared to their counterparts in the normal group [33]. Correspondingly, a British cohort study by Atkins et al., involving 4252 participants, revealed a significant elevation in systolic and diastolic blood pressure in sarcopenia than healthy population [45]. Androga and colleague [44] demonstrated that the sarcopenia group had increased prevalence of blood pressure, compared to the healthy counterparts.

The increase in SBP and DBP in individuals with sarcopenia can be attributed to skeletal muscle loss resulting from metabolic alterations and a decline in muscle mass. This phenomenon contributes to diminished energy expenditure, decreased physical activity, insulin resistance, and heightened arterial stiffness in older adults. Additionally, the accumulation of excessive visceral fat may induce an inflammatory response, leading to the thickening of blood vessel walls, constriction of vascular passages, and hindrance of blood flow. Hence, it is imperative to emphasize the potential health implications of elevated SBP and DBP in individuals with sarcopenia. These repercussions include reduced energy expenditure and physical activity, heightened susceptibility to insulin resistance, and an increased likelihood of arterial stiffness among the elderly. Furthermore, the accrual of surplus visceral fat mass has emerged as a pivotal factor in triggering an inflammatory response, subsequently fostering structural changes in blood vessels, constricting vascular passages, and impeding blood flow.

The current study demonstrated a notable strength by focusing on the investigation of risk factors, specifically among females within the representative population of late middle-aged individuals. This age group is particularly significant because sarcopenia progresses rapidly and complications begin in this age group [46,47,48,49]. These findings offer valuable insights into the early detection and treatment of sarcopenia. However, it is crucial to acknowledge several limitations of the present study that should be addressed in future research. First, despite the inclusion of a substantial sample size of 2814 participants with representative statistical weights, the use of a cross-sectional design may have restricted the ability to establish causal relationships for the identified risk factors. Factors such as elevated triglyceride and total cholesterol levels have been implicated as potential predictors of sarcopenia. The cross-sectional nature of the study raises the possibility that sarcopenia itself could influence the blood test results. Thus, further research is imperative to comprehensively elucidate the intricate relationship between these predictors and the development of sarcopenia. Future studies could enhance the robustness of their findings by considering longitudinal or randomized case-control designs. Another limitation was the omission of an examination of sarcopenic obesity, a condition characterized by low muscle mass and high body fat. The absence of consideration for sarcopenic obesity is particularly relevant, as it may influence alterations in total cholesterol and triglyceride levels. To facilitate a more nuanced interpretation of the study results, future research should consider the potential impact of sarcopenic obesity on the identified metabolic parameters. In addition, the present study did not investigate the assessment of protein intake. Protein intake is essential for the prevention and intervention of sarcopenia. Furthermore, the diagnosis of sarcopenia did not take into account muscle strength and function. If these factors had been considered, better results could have been achieved. Finally, the study did not consider variables from psychosocial aspects. This should be considered in the next study.

## 5. Conclusions

The current nationwide investigation provides the first clinical findings of risk factors and the prevalence of sarcopenia in late middle-aged women.

Within this specific demographic, the prevalence of sarcopenia was estimated to be 13.87%, accompanied by a confidence interval spanning from 12.15% to 15.78%. The study identified clinical risk factors associated with sarcopenia, encompassing parameters such as waist circumference, systolic and diastolic blood pressure, as well as triglyceride and total cholesterol levels. By acknowledging both the prevalence and identified risk factors, healthcare professionals might have an improved capacity to identify and detect potential cases of sarcopenia among female patients. However, further research is required to deepen our understanding of the relationship between these risk factors and sarcopenia and to bolster the robustness of these findings. Exploring longitudinal or randomized case-control study designs holds promise for unraveling the intricacies of this association.

## Figures and Tables

**Figure 1 healthcare-12-00362-f001:**
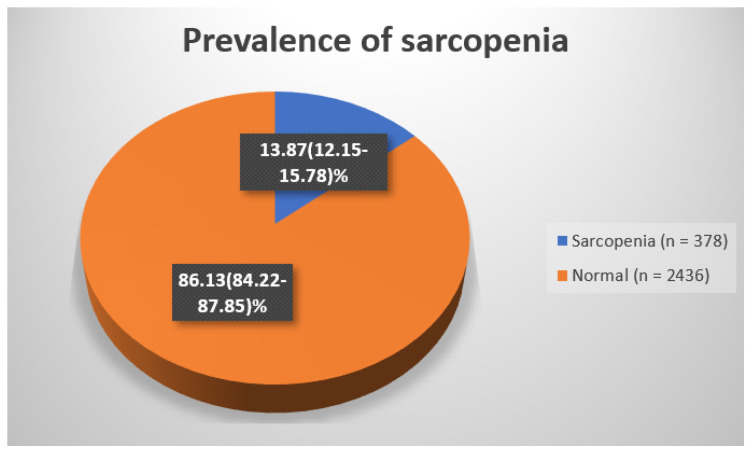
Prevalence of sarcopenia.

**Table 1 healthcare-12-00362-t001:** Prevalence of sarcopenia.

	Sarcopenia	Normal	Total
	(*n* = 378)	(*n* = 2436)	(N = 2814)
Un-weighted (%)	13.43	86.57	100
Weighted (%)	13.87 (12.15–15.78)	86.13 (84.22–87.85)	100

Weighed values present the 95% confidence interval.

**Table 2 healthcare-12-00362-t002:** Clinical risk factors for sarcopenia.

	Sarcopenia	Normal	*p*
	(*n* = 378)	(*n* = 2436)	
Age (years)	57.648 ± 4.164	56.301 ± 4.366	0.000
Height (cm)	149.69 ± 4.732	156.177 ± 4.748	0.000
Weight (kg)	58.954 ± 9.042	58.499 ± 7.884	0.307
BMI (kg/m^2^)	26.265 ± 3.537	23.961 ± 2.854	0.000
WC (cm)	85.635 ± 9.675	81.171 ± 8.575	0.000
SMI (kg/m^2^)	485.397 ± 29.216	617.959 ± 62.806	0.000
FG (mg/dL)	100.85 ± 23.259	98.956 ± 22.479	0.138
Triglyceride (mg/dL)	145.047 ± 78.063	128.908 ± 85.84	0.001
TC (mg/dL)	207.529 ± 36.083	200.999 ± 36.093	0.001
SBP (mmHg)	128.726 ± 17.94	123.392 ± 17.574	0.000
DBP (mmHg)	80.242 ± 9.501	78.406 ± 10.472	0.001
Drinking status (%) (current-/ex-/non-user)	58.262/19.83/21.908	58.724/15.851/25.425	0.156
Smoking status (%) (current-/ex-/non-user)	4.764/2.849/92.387	5.943/1.228/92.83	0.121

Values were presented as the mean with accompanying standard deviation. The statistical analyses employed included the independent *t*-test and the chi-square test. BMI, body mass index; WC, waist circumference; SMI, skeletal muscle mass index; SBP, systolic blood pressure; DBP, diastolic blood pressure; FG, fasting glucose; and TC, total cholesterol.

**Table 3 healthcare-12-00362-t003:** Multiple logistic regression for odds ratios of sarcopenia.

Variables	Odd Ratios (95% of CI)	*p*
BMI	19.178 (4.3080–85.380)	0.015
WC	1.362 (1.011–1.836)	0.042
SMI	0.115 (0.067–0.193)	0.000
SBP	1.349 (1.255–1.451)	0.000
DBP	1.845 (1.646–2.067)	0.000
TC	1.166 (1.141–1.191)	0.000
Triglyceride	1.078 (1.058–1.097)	0.000

The 95% confidence interval for the odds ratio was determined through the application of multiple logistic regression. BMI, body mass index; WC, waist circumference; SMI, skeletal muscle mass index; SBP, systolic blood pressure; DBP, diastolic blood pressure; and TC, total cholesterol.

## Data Availability

All data were anonymized and can be downloaded from the website at https://knhanes.kdca.go.kr/knhanes, accessed on 1 December 2023.

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
