# Peer review of "Korean Nationwide Exploration of Sarcopenia Prevalence and Risk Factors in Late Middle-Aged Women"

_healthcare, 2024, doi:10.3390/healthcare12030362_

Round 1
Reviewer 1 Report
Comments and Suggestions for Authors
The title "Nationwide Exploration of Sarcopenia Prevalence and Risk Factors in Late Middle-Aged Women" effectively communicates the focus and scope of the study. However, it could be improved by providing a more specific indication of the nation in which the study was conducted, as "nationwide" could refer to any country. Additionally, including the specific age range of the late middle-aged women (50-64 years old) in the title would make it more informative.
As for the abstract, it could benefit from a clearer indication of the significance of the findings and how they contribute to the existing literature on sarcopenia. Additionally, the abstract could be improved by briefly mentioning the methodology used to analyze the data, providing readers with a better understanding of the study's approach.
For introduction, It could benefit from a more structured approach. It jumps from discussing the definition and consequences of sarcopenia to the prevalence in Korea and Asia, and then to the challenges faced by healthcare professionals in diagnosing sarcopenia. A more organized flow of information would make it easier for readers to follow. I recommended to focus more on litearature of sarcopenia: the following reference is recommended:
Siahpoosh, Marzieh Beygom, B. E. N. Helmi, and Soheil Gholibeygi. "Avicenna's Views on Lifestyle Recommendations for the Elderly: Strategies to Address Age-Related Sarcopenia." Health Nexus 1.1 (2023): 1-3.
Apparently, there are many factors contributing to middle-aged health as risk factorm hence, you should consider them more inclusively in different aspects such as psychologically, physiologically, physically,.... the following references are helpful.
Park, Sam-Ho, et al. "The Effects of Lumbar Stabilization Exercise Program Using Respiratory Resistance on Pain, Dysfunction, Psychosocial Factor, Respiratory Pressure in Female Patients in’40s with Low Back Pain: Randomized Controlled Trial." Annals of Applied Sport Science 9.3 (2021): 0-0.
Methode
The study excluded a significant number of individuals (34,123 out of 37,753) based on age and gender criteria, which may have introduced selection bias. It's important to justify the exclusion criteria and consider how this might impact the generalizability of the findings.
The description of data collection and measurement procedures is thorough, but it would be beneficial to include information on the reliability and validity of the measurement tools used. Additionally, details about the inter-rater reliability and calibration procedures for measurements would enhance the method's transparency.
While the method mentions approval by the Disease Control and Prevention Center Ethics Review Board and participants' informed consent, it would be valuable to include more details about ethical considerations, such as confidentiality, participant anonymity, and potential conflicts of interest.
For the result section, using figures for showing the findings would make more undrstanding for readers.
Discussion
This part is well designed and written.
Author Response
Response to the reviewers’ comments
At first, authors express their deep gratitude to the reviewer’s valuable comments. know it is arduous job for reviewing papers. It consumes lots of time and effort. We were able to learn the way to write paper correctly because of your delicate comments. And we came to realize citing new references is very important, again. We really appreciate it with my whole heart.
*Please find attached our revised manuscript with changed from the original version highlighted with green (Please click an author's note file).
- The title "Nationwide Exploration of Sarcopenia Prevalence and Risk Factors in Late Middle-Aged Women" effectively communicates the focus and scope of the study. However, it could be improved by providing a more specific indication of the nation in which the study was conducted, as "nationwide" could refer to any country. Additionally, including the specific age range of the late middle-aged women (50-64 years old) in the title would make it more informative.
Authors response: Thank you very much for your suggestion, Authors agree with the reviewer’s comment. And We put the specific nation. (line 2, page 1).
However, we did not include specific age range this is because we already in put the age range in abstract and introduction and method. Authors feel sorry for not following all the whole of your comments.
- As for the abstract, it could benefit from a clearer indication of the significance of the findings and how they contribute to the existing literature on sarcopenia.
Authors response: Authors totally agree with the reviewer’s comment. We I wrote about the effects of the paper and how it would be beneficial for healthcare profession. (lines 27-29, page 1).
- Additionally, the abstract could be improved by briefly mentioning the methodology used to analyze the data, providing readers with a better understanding of the study's approach.
Authors response: Authors entirely agree with the reviewer’s comment. However HealthCare journal limits the abstract to 200 words, preventing further content additions. We are not able to add any words. The abstract in this manuscript has already reached the 200-word limit.
- For introduction, It could benefit from a more structured approach. It jumps from discussing the definition and consequences of sarcopenia to the prevalence in Korea and Asia, and then to the challenges faced by healthcare professionals in diagnosing sarcopenia. A more organized flow of information would make it easier for readers to follow. I recommended to focus more on litearature of sarcopenia: the following reference is recommended:
Siahpoosh, Marzieh Beygom, B. E. N. Helmi, and Soheil Gholibeygi. "Avicenna's Views on Lifestyle Recommendations for the Elderly: Strategies to Address Age-Related Sarcopenia." Health Nexus 1.1 (2023): 1-3.
Authors response: Author response: Authors totally agree with the reviewer’s comment. We reorganized it. These following things are logical flow of idea (pages 1-2).
[Sarcopenia definition & body composition]
Sarcopenia is characterized by age-related reductions in skeletal
muscle mass, resulting in diminished muscle strength, function, and quality of life [1]. Although various studies have indicated that hormonal changes, immobility, age-related changes in muscle composition, nutritional factors, and neurodegenerative processes contribute to its development, the precise mechanism of sarcopenia is not yet fully understood. Importantly, sarcopenia is particularly common among individuals aged 65 years and above [2].
[Importance of sarcopenia research in Asia]
The elderly population in Asia is rapidly growing, and Korea is among the countries with the highest rates of population aging worldwide. As of 2021, approximately 16.5% of the Korean population is aged 65 years or older, and this percentage is expected to increase to approximately 40% by 2050 [3]. Sarcopenia is more prevalent in Korea and Asia than in other countries.
[High prevalence of female than male]
Several studies have reported a higher prevalence of sarcopenia in females compared to males. In a screen of 10,063 individuals, Dam et al. announced that a prevalence of 11.80% in females and 5.10% in males [4]. Similarly, Hunt et al. examined about two thousand community dwelling Japanese old population and found a sarcopenia prevalence of 16.56% in females and 10.34% in males [5]. A significant proportion of the elderly population in Korea, particularly women, is vulnerable to sarcopenia.
[However, insufficient female research of sarcopenia aged 50-64 and its importance]
However, compared with the extensive research on sarcopenia in males, the early detection of sarcopenia in females remains a challenging task [6-9]. Healthcare professionals, encompassing physical therapists and primary care clinicians, face difficulties in diagnosing sarcopenia owing to their limited knowledge and diagnostic tools despite the potential negative consequences associated with sarcopenia and the growing population of elderly females. In primary clinical settings, clinicians must assess the likelihood of sarcopenia before considering referrals for diagnosis and treatment. Moreover, the lack of awareness among clinicians regarding sarcopenia as a distinct disease increases the risk of missed diagnoses [10]. Therefore, it is crucial for healthcare professionals and primary clinicians to understand the characteristics of key risk factors associated with early detection and prevention to effectively address this challenge [11]. Prompt identification and early detection of individuals manifesting symptoms indicative of sarcopenia are essential to ensure timely diagnosis and intervention. Failure to diagnose the condition promptly can result in complications, including compromised functional recovery, diminished quality of life, and inefficient utilization of healthcare resources.
[Research purpose and hypothesises]
However, the majority of studies on sarcopenia have primarily focused on individuals aged 65 years and older [12-15], even though age-related muscle loss can begin as early as the 50s [16-20]. It is crucial to identify the risk factors for muscle loss at an earlier stage to effectively prevent and treat this condition. Therefore, this study aimed to investigate the prevalence of sarcopenia and its associated risk factors in women aged 50 to 64 years. We hypothesized that this age group would have specific risk factors and prevalence rates that would differ from those observed in older individuals.
- Apparently, there are many factors contributing to middle-aged health as risk factorm hence, you should consider them more inclusively in different aspects such as psychologically, physiologically, physically,.... the following references are helpful.
Park, Sam-Ho, et al. "The Effects of Lumbar Stabilization Exercise Program Using Respiratory Resistance on Pain, Dysfunction, Psychosocial Factor, Respiratory Pressure in Female Patients in’40s with Low Back Pain: Randomized Controlled Trial." Annals of Applied Sport Science 9.3 (2021): 0-0.
Authors response: Authors fully agree with the reviewer’s comment. Authors feel sorry for missing psyco-social aspect variables. This study did not consider variables from psychosocial aspects. This should be considered in the next study. We mentioned on the limitation part (lines 497-498, page 12)
- (Method) The study excluded a significant number of individuals (34,123 out of 37,753) based on age and gender criteria, which may have introduced selection bias. It's important to justify the exclusion criteria and consider how this might impact the generalizability of the findings.
Authors response: Thank you very much for your suggestion. Authors have contemplated the reviewer’s comment and we would like to give the following explanation:
The results pertain specifically to late middle-aged women ranged 50-64, and therefore, generalization to other age groups is not warranted. This study targeted a population within this age range, and it cannot be extrapolated to different age groups. In Korean National Health and Nutrition Examination Survey, the participants were randomly selected from the population, ensuring the absence of selection bias in this study.
The authors apologize for any confusion caused to the reviewer.
- The description of data collection and measurement procedures is thorough, but it would be beneficial to include information on the reliability and validity of the measurement tools used. Additionally, details about the inter-rater reliability and calibration procedures for measurements would enhance the method's transparency.
Authors response: Authors entirely agree with the reviewer’s comment. We added it (line 277, page 5)
- While the method mentions approval by the Disease Control and Prevention Center Ethics Review Board and participants' informed consent, it would be valuable to include more details about ethical considerations, such as confidentiality, participant anonymity, and potential conflicts of interest.
Authors response: Authors express their deep gratitude to the reviewer’s valuable comment. We added it (lines 97-102, page 3).
- For the result section, using figures for showing the findings would make more understanding for readers.
Authors response: Authors totally agree with the reviewer’s comment. We add a figure (line 261, page 4)
Authors sincerely express their gratitude to the reviewer’s valuable comments, which have improved the readability and quality of this paper a lot.

Reviewer 2 Report
Comments and Suggestions for Authors
I consider the theme of this article to be relevant: “National exploration of the prevalence of sarcopenia and risk factors in late middle-aged women;”
I consider this article to be relevant, because sarcopenia is a very prevalent pathology that can be counteracted, so more and better scientific evidence is needed in this area of knowledge;
I therefore make some suggestions that I believe could be a contribution to clarifying and enriching the article:
I would like to have seen a more robust theoretical framework for the reader to better understand what sarcopenia is, how the diagnosis is made and the evolution of knowledge on this topic.
Introduction
Paragraph 25
…. is a progressive and generalized skeletal muscle disorder… It is a characterized disease…..!
Section 42
…… A significant proportion of the elderly population in Korea, especially women, are vulnerable to sarcopenia. ….Why ?
Methods
It is not clear how the diagnosis of sarcopenia was made, did they only use lean muscle mass? and not the Force?
Section 77
…..Participants were divided into two groups based on sarcopenia criteria, with 378 individuals in the sarcopenia group and 2,436 in the normal group…… What were the criteria?
Alinea 101
………implied a determination of the SMI value below 0.521 in women, as per the guidelines………..These guidelines[21]. are old, hear an update on these criteria: Cruz Jentoft 2019, Reference 16. And was strength not considered when making the diagnosis of sarcopenia?
Alinea 128
………..weight variables, fasting state….I don’t understand the fasting state!
Discussion
Section 293
……. To facilitate a more nuanced interpretation of study results, future research should consider the potential impact of sarcopenic obesity on identified metabolic parameters…… As well as assessment of protein intake!!!!!
Author Response
At first, authors express their deep gratitude to the reviewer’s valuable comments. know it is arduous job for reviewing papers. It consumes lots of time and effort. We were able to learn the way to write paper correctly because of your delicate comments. And we came to realize citing new references is very important, again. We really appreciate it with my whole heart.
*Please find attached our revised manuscript with changed from the original version highlighted with yellow (Please click an author's note file).
- I consider the theme of this article to be relevant: “National exploration of the prevalence of sarcopenia and risk factors in late middle-aged women;”
Authors response: Authors totally agree with the reviewer’s comment. We partially changed it (lines 1-4, page 1)
I consider this article to be relevant, because sarcopenia is a very prevalent pathology that can be counteracted, so more and better scientific evidence is needed in this area of knowledge; I therefore make some suggestions that I believe could be a contribution to clarifying and enriching the article: I would like to have seen a more robust theoretical framework for the reader to better understand what sarcopenia is, how the diagnosis is made and the evolution of knowledge on this topic.
Introduction
- (Paragraph 44)…. is a progressive and generalized skeletal muscle disorder… It is a characterized disease…..!
Authors response: Authors entirely agree with the reviewer’s comment. We I wrote about the effects of the paper and how it would be beneficial for healthcare profession. (lines 41-43, page 1).
- (Section 55)…… A significant proportion of the elderly population in Korea, especially women, are vulnerable to sarcopenia. ….Why ?
Authors response: Authors feel sorry for missing the clear explanation.
The explanation immediately follows. Although the prevalence is high, there is limited existing research, emphasizing the vulnerability of women. To prevent any misunderstandings, the manuscript has been revised accordingly (line 61-63, page 1).
Authors feel sorry for misleading you by missing the clear explanation. We apologized to confuse you, again.
- (Methods) It is not clear how the diagnosis of sarcopenia was made, did they only use lean muscle mass? and not the Force?
Authors feel sorry for missing additional detail of muscle power and function in diagnosis of sarcopenia in the original manuscript. Therefore, We address in research limitations section (lines 494-497, page 10).
Thank you very much for your corrections.
- (Section 77) …..Participants were divided into two groups based on sarcopenia criteria, with 378 individuals in the sarcopenia group and 2,436 in the normal group…… What were the criteria?
Authors response: Authors feel sorry for missing detail explanation of the criteria in the original manuscript. We added it (lines 103-109, page 3).
- (line 101)………implied a determination of the SMI value below 0.521 in women, as per the guidelines………..These guidelines[21]. are old, hear an update on these criteria: Cruz Jentoft 2019, Reference 16. And was strength not considered when making the diagnosis of sarcopenia?
Authors response: Authors feel sorry for missing additional detail of muscle in diagnosis of sarcopenia in the original manuscript. Therefore, this has been addressed in research limitations section (lines 494-497, page 12).
Next time we promise to add the muscle power in diagnosis of sarcopenia in the future research
Thank you very much for your specific suggestion.
- (linea 128) ………..weight variables, fasting state….I don’t understand the fasting state!
Authors response: We apologize for making you confused. We are trying to find the words ‘fasting state’, unfortunately we were not able to find it. If you meaning ‘Fasting glucose’, asting glucose refers to the concentration of glucose measured in the blood after an overnight fast. Simply put, it represents the blood sugar level measured in the morning before consuming any food or drinks. Fasting glucose is commonly used in the medical and health fields to assess conditions such as diabetes or glycated hemoglobin. Typically, normal fasting glucose falls within the range of 70-100 mg/dL. We provided more information (lines 130-141, pages 3-4).
Authors feel sorry for misleading you by missing the clear explanation. We apologized to confuse you, again.
- (Discussion) (Section 293) ……. To facilitate a more nuanced interpretation of study results, future research should consider the potential impact of sarcopenic obesity on identified metabolic parameters…… As well as assessment of protein intake!!!!!
Authors response: Authors fully agree with the reviewer’s comment. Authors feel sorry for missing Assessment of protein intake. Therefore, We addressd in research limitations section (lines 485-497, pages 11-12)(lines 492-494, page 12).
Next time we promise to add the muscle power in diagnosis of sarcopenia and assessment of protein intake and other various factors in the future research.
Thank you very much for your specific suggestion.
Thank you very much for the reviewer’s so many valuable comments. Thanks to the reviewer, the authors have improved their understanding of this research contents and the quality of the paper a lot. Once again, thank you.
Round 2
Reviewer 1 Report
Comments and Suggestions for Authors
Accepted